# Bioabsorbable Magnesium-Based Materials Potential and Safety in Bone Surgery: A Systematic Review

**DOI:** 10.3390/cmtr18020024

**Published:** 2025-04-07

**Authors:** Chun Ho Hung, Yui Chit Kwok, Jason Yip, Ho Hin Wong, Yiu Yan Leung

**Affiliations:** Oral and Maxillofacial Surgery, Faculty of Dentistry, The University of Hong Kong, Hong Kong, China; leohung@connect.hku.hk (C.H.H.); walterk6@connect.hku.hk (Y.C.K.); u3571854@connect.hku.hk (J.Y.); jackyhhw@connect.hku.hk (H.H.W.)

**Keywords:** bone surgery, absorbable, magnesium, osteosynthesis, fixation

## Abstract

The goal of this study was to evaluate the clinical outcomes, safety, and clinical applications of bioabsorbable magnesium-based materials for fixation in bone surgeries. The review was conducted based on the Preferred Reporting Items for Systematic Reviews and Meta-Analyses (PRISMA) statement. An initial search was performed on electronic databases, followed by manual and reference searches. The articles selected were evaluated for patient characteristics, biocompatibility, the need for revision surgery, bone union rates, and the incidence of gas formation associated with implant degradation. Out of the 631 initially identified articles, 8 studies including a total of 386 patients were included in the final qualitative analysis. The magnesium (Mg) group carried a lower rate of revision surgery (1/275) when compared to the titanium (Ti) group (18/111). A high rate of bone union was found in the Mg group and a low infection rate (3/275) was found in the Mg group. The serum level of Mg and calcium (Ca) were not found to be affected. Mg implants are applied in various orthopedic surgeries but they are not applied in in oral or maxillofacial surgeries. Mg implants appear to be a safe alternative for bone fixation and are resorbable. Future research into the application of Mg implants in bone fixation in different anatomical sites is essential to fully harness their potential benefits for patients.

## 1. Introduction

Bioabsorbable materials are used in a variety of orthopedic and craniofacial surgeries, designed to gradually degrade and be metabolized by the physiological environment, and ultimately eliminating the need for the secondary removal of osteosynthesis hardware. The most widely studied in vitro and in vivo bioabsorbable polymers are Poly-lactides, which include Poly(L-lactic acid) (PLLA), Poly(D-lactic acid) (PDLA), and Poly(lactide-co-glycolide) (PLGA) [1]. They were found to produce an acidic environment during degradation which is unfavorable for tissue healing, and they have low mechanical strength [2]. Therefore, there is a growing interest in bioabsorbable metals, such as magnesium (Mg) and its alloys, which combine mechanical strength with gradual degradation, aligning with the natural healing process of bone and soft tissues.

In modern surgical practices, conventional non-resorbable metal implants are extensively used due to their high mechanical strength, biocompatibility, and established clinical efficacy, with titanium (Ti) being the most popular choice. In terms of mechanical properties, in vitro studies have demonstrated that the modulus of elasticity of Mg (44 GPa) is closer to that of natural human bone (5–23 GPa) [2], while the modulus of elasticity of Ti is 110 GPa [3]. Although Ti implants have high mechanical strength that could provide stable fixation, the high Young’s modulus causes stress shielding, a condition that may lead to bone resorption [4]. Moreover, the use of conventional implants may cause complications such as unacceptable palpability, plate exposure, infection, and allergic hypersensitivity, which increases treatment costs for a secondary surgery to remove the implants and adds to patients’ psychological burden [5].

Mg metal and its alloys are considered to be promising candidates for resorbable implants. The first human study involving Mg implants was performed by Erwin in 1906 [6]. Common examples of Mg alloys include the “AZ” series, which consists of magnesium, aluminum, and zinc; the “WE” series, which consists of magnesium and rare earth elements such as neodymium; the “ZK” series, which consists of magnesium, zinc, and zirconium [7]. Major benefits of Mg alloys as compared to pure Mg include improved mechanical strength, a controlled degradation rate, and enhanced corrosion resistance [6]. The self-degradation mechanism of Mg implants is triggered upon contact with body fluids, which undergo corrosion to form magnesium hydroxide (Mg(OH)_2_) and hydrogen gas. Most degradation products will be eliminated by the kidney or liver, while a small portion of them will be utilized by other tissues [8]. Mg is also biocompatible and can promote angiogenesis and, hence, bone healing, as has been shown in in vitro and animal studies [9]. Bone surgeries would therefore be advantageous if bioresorbable magnesium implants were proven to be safe and effective. Currently, Mg metal and its alloys are gaining popularity in orthopedic fields, such as hallux valgus chevron osteotomy, proximal tibia, distal femur, femoral neck, etc. However, as Mg and its alloys are relatively new in these applications, they are not yet widely used in view of the unknown short- and long-term risks, biocompatibility, and clinical outcomes. Individual smaller-scale studies on their applications are emerging, but there is still a lack of large-scale studies or clear evidence on their clinical performances and applications in the field of bone fixation. It is unknown whether the use of Mg and its alloys may be applicable in the field of craniomaxillofacial surgery.

Therefore, the aim of this systematic review is to evaluate the clinical outcomes, safety, and applications of Mg-based osteosynthesis material.

## 2. Materials and Methods

### 2.1. Study Design

To provide the best evidence regarding the application of Mg in human bone fixation, a systematic review was conducted according to the Preferred Reporting Items for Systematic Reviews and Meta-Analyses (PRISMA 2020) guidelines [10]. The systematic review is registered in the Open Science Framework https://doi.org/10.17605/OSF.IO/GEZMJ (accessed on 2 July 2024).

### 2.2. Study Selection

The systematic review was structured using the PICO framework: population (P)—studies involving human patients with fractures requiring surgical fixation with screws were included; intervention (I)—studies that utilized Mg screws for bone fixation were considered; control (C)—comparative studies involving patients who underwent bone fixation with conventional titanium (Ti) screws, which is the current standard in orthopedic surgery, were included; outcome (O)—studies had to report clinical and radiographic outcomes following bone fixation in order to be included. The outcomes assessed included biocompatibility, the need for revision surgery, bone union rates, and the incidence of gas formation associated with implant degradation.

### 2.3. Search Strategy

Two rounds of comprehensive searches and one round of evaluation regarding the application of Mg in bone fixation were conducted by independent reviewers. The titles and abstracts of all the retrieved articles were screened. When there were disputes or inconsistencies, consensus was obtained by screening the full text of the involved articles. No automation tools were integrated within the overall study selection process. The method of analysis, inclusion criteria, and exclusion criteria for the review are specified below and documented in a protocol.

#### 2.3.1. First Round Search

The first round of searches was conducted on electronic databases, including PubMed, Medline, and the Cochrane Library, all with the following keywords: [“Magnesium”] AND [“Plate” OR “Fixation” OR “Implant”] AND [“Orthopaedic” OR “Bone” OR “Maxillofacial” OR “Orthognathic” OR “Trauma”]. No limits were applied for the language; foreign-language papers were translated if necessary. No limits were applied for publication status. The final date of the search was 31 December 2023. Articles that were related to the application of magnesium in human bone fixation were selected for the second round of searches. The publication dates were limited to within 10 years. A “Human” filter was applied for the initial search. The titles and abstracts of the articles were screened. For studies that appeared to be relevant, but which had insufficient detail in the abstracts, full-article texts were retrieved for further evaluation.

#### 2.3.2. Second Round Search

The second round of searches consisted of reference list searches from selected articles from the first round of searches. Reference list searches were performed by independent reviewers. Articles included from the first and second rounds of searches were entered into the third round.

#### 2.3.3. Third Round Search

Articles from the first and second rounds of searches were evaluated against four criteria: (i) the study was a clinical study on human subjects; (ii) the study consisted of a sample size with at least 20 subjects; (iii) the study reported on the duration of subject follow-up, and we required at least 1 year of follow-up; (iv) the study involved the application of Mg in bone fixation with implant/screw/plate; (v) the study reported on the clinical and radiographic treatment outcomes. Articles that satisfied all five criteria were selected for the final review.

### 2.4. Data Extraction

Studies that fulfilled the predefined inclusion criteria were incorporated into the final review for data extraction and subsequent analyses. The data extracted from the included studies were as follows: (i) participant characteristics (number of participants, type of bone injuries, duration of participation); (ii) type of study; (iii) inclusion and exclusion criteria; (iv) details of surgical procedure and how application of magnesium was applied in bone fixation; (v) clinical outcome measures (infection rate, serum level, rate of revision surgery, and rate of Mg implant degradation); (vi) radiographic outcome measures (bone union and radiolucent zone); (vii) follow-up period and follow-up measures of Mg bone fixation.

## 3. Results

A total of 631 articles were identified from the electronic databases, all of which discussed the application of Mg in bone fixation. A total of 211 duplicated papers were identified and removed manually. After the initial screening, 49 papers were further removed.

The reference list and manual searches contributed 9 articles. A total of 58 articles were evaluated against the five criteria for inclusion, and 50 articles were excluded. Finally, eight articles were found to be eligible for the final review. A flow diagram of the article selection process is presented in Figure 1.

A total of eight full-text articles were identified based on the inclusion criteria. The risk of bias in the included studies was assessed following the seven domains. Out of the eight included articles, five were considered to have a low risk of bias, and three were considered to have a moderate risk of bias. None of the studies were classified to have a high risk of bias (Table 1).

The eight selected articles focused on the application of Mg in human bone fixation and these were selected for qualitative analysis. The analysis included a cohort of 386 subjects from various studies. The study designs included one randomized clinical trial (RCT), one cohort study, and six case series studies (Table 2).

To evaluate the risk of bias in the included studies, an assessment was developed based on the ROBINS-I tool [11]. The ROBINS-I tool utilized seven domains to evaluate risk of bias in the included studies: (i) bias due to confounding; (ii) bias in the selection of participants for the study; (iii) bias in the classification of the interventions; (iv) bias due to deviations from the intended interventions; (v) bias due to missing data; (vi) bias in the measurement of the outcomes; (vii) bias in the selection of the reported results (Table 1). The quality of each study was assessed independently by reviewers, and any discrepancies were resolved through discussion.

A study that meets all seven domains would be categorized as having a “low potential risk of bias”, whereas a study meeting four or more criteria would be considered as having a “moderate potential risk of bias”. Studies fulfilling three or fewer criteria would be classified as having a “high potential risk of bias”. Studies with a “low” or “moderate risk of bias” were included in the following data analysis.

**Table 1 cmtr-18-00024-t001:** Risk of bias assessment of included studies.

Studies	D1	D2	D3	D4	D5	D6	D7	Overall
Stürznickel J et al. [12]	No	No	No	No	No	No	No	Low
May H et al. [13]	No	No	No	No	Yes	Yes	No	Moderate
Zhao D et al. [8]	No	No	No	No	No	No	No	Low
Lee JW et al. [14]	No	No	No	No	Yes	Yes	No	Moderate
Polat O et al. [15]	No	No	No	No	No	No	No	Low
Herber V et al. [16]	No	No	No	No	No	No	No	Low
Choo JT et al. [17]	No	No	No	No	Yes	Yes	No	Moderate
Lee CH et al. [18]	No	No	No	No	Yes	Yes	No	Moderate

D1—bias due to confounding; D2—bias in selection of participants in the study; D3—bias in classification of interventions; D4—bias due to deviations from intended interventions; D5—bias due to missing data; D6—bias in measurement of the outcomes; D7—bias in the selection of the reported result.

The patient mean age across the studies varied significantly, reflecting a wide range of applications for Mg-based implants in different age groups. The youngest cohort was in the study by Stürzknick et al. [12], which focused on children and adolescents, with a mean age of 12.9 ± 3.2 years. On the contrary, Choo et al. [17] included patients with a mean age of 54.5 ± 12.0 years, targeting a predominantly older population undergoing hallux valgus surgery.

**Table 2 cmtr-18-00024-t002:** Summary of the studies in the final review.

Authors	Year	Title	Study Design	Number of Participants	Patient Mean Age	Anatomical Site	Type of Mg Used	Titanium Screws Use	Outcome Measures	Follow-Up Time Point	Country
Stürznickel et al. [12]	2021	Safety and performance of biodegradable magnesium-based implants in children and adolescents	Case series	Mg: 89	Mg: 12.9 ± 3.2	Proximal tibia Elbow Upper Ankle joint Patella Distal fermur	MgYREZr	/	Combination of clinical (range of motion, functions, pain) and radiographic (X-rays and/or MRI) findings.	4th–8th weeks, 3rd–6th months	Germany
May et al. [13]	2020	Bioabsorbable magnesium screw versus conventional titanium screw fixation for medial malleolar fractures	Case series	Mg: 23 Ti: 25	Mg: 37.9 ± 17.7 Ti: 45 ± 15.7	Medial malleolar	MgYREZr	Yes	Combination of clinical (infection, wound problem, ankle instability, tendon subluxation, AOFAS, LK grading), and radiographic (X-rays and CT) findings.	Mean 24.7 ± 12.0 months (no scheduled follow-up due to retrospective in nature)	Turkey
Zhao et al. [8]	2015	Vascularized bone grafting fixed by biodegradable magnesium screw for treating osteonecrosis of the femoral head	RCT	Mg: 23	Mg: 30 ± 7	Femoral head	Pure Mg (99.99 wt.%)	/	Combination of clinical (Harris hip score) and radiographic (X-rays and CT) findings and blood test (trace element detector for serum level).	1st, 3rd, 6th, and 12th month	China
Lee JW et al. [14]	2016	Long-term clinical study and multiscale analysis of in vivo biodegradation mechanism of Mg alloy	Case series	Mg: 53	N/A	Scaphoid, distal radius	5 wt% Ca, 1 wt% Zn screw	/	Combination of clinical (VAS, DASH, range of motion, grip power) and radiographic (X-rays) findings.	1st and 2nd week, 1st, 2nd, 3rd, and 6th month, 1st year	Republic of Korea
Polat et al. [15]	2021	Surgical outcomes of scaphoid fracture osteosynthesis with magnesium screws	Case series	Mg: 21	Mg: 28.5 ± 5.8	Scaphoid	MgYREZr	/	Combination of clinical (grip strength, pinch strength, and range of motion) and radiographic (X-rays) findings.	Mean: 43.3 ± 5.3 months (no scheduled follow-up due to retrospective nature of study)	Turkey
Herber et al. [16]	2021	Can Hardware Removal be Avoided Using Bioresorbable Mg-Zn-Ca Screws After Medial Malleolar Fracture Fixation? Mid-Term Results of a First-In-Human Study	Non-randomized cohort study	Mg: 20 Ti: 17	Mg: 40.1 ±14.5 Ti: N/A	Medial malleolar	ZX00 rods (99.1 wt% Mg, 0.45 wt% Zn, and 0.45 wt% Ca)	Yes	Combination of clinical (VAS, AOFAS, blood analysis, and range of motion) and radiographic (X-rays) findings.	6th and 12th month	Austria
Choo et al. [17]	2018	Magnesium-based bioabsorbable screw fixation for hallux valgus surgery—A suitable alternative to metallic implants	Case series	Mg: 24 Ti: 69	Mg: 54.5 ± 12.0 Ti: N/A	Hallux Vagus	MgYREZr	Yes	Combination of short form 36 (SF-36), AOFAS-HMI, VAS, and radiological evaluation (pre-op, 3 and 12 months); limited field forefoot CT scan was performed at 12 months.	3rd and 12th month	Singapore
Lee CH et al. [18]	2023	Results of the Use of Bioabsorbable Magnesium Screws for Surgical Treatment of Mason Type II Radial Head Fractures	Case series	Mg: 22	Mg: 52.3	Radial head	Mg Screw (Resomet: U&I Corp Seoul, Republic of Korea)	/	Combination of disabilities of the arm, shoulder, and hand (DASH) score, mayo elbow performance score (MEPS), range of joint motion and hand grip power, measured at 6 months post-op; radiographic tests performed at 2 weeks, 4 weeks, 8 weeks, 12 weeks, 6 months, and 1 year.	2nd, 4th, 8th, 12th, and 6th month, 1st year	Republic of Korea

### 3.1. Types of Magnesium Used

There were a total of 275 patients who received Mg-based devices for bone fixation from the included studies. The Mg-based devices include MgYREZr (manufactured by MAGNEZIX^®^), 99.99 wt.% Mg (manufactured by Dongguan Eontec Co., Ltd., Dongguan, China), ZX00 rods (99.1 wt% Mg, 0.45 wt% Zn, and 0.45 wt% Ca), Mg screws manufactured by Resomet, Korea (composition not specified), and Mg screws (5 wt% Ca, 1 wt% Zn) [8,12,13,14,15,16,17,18].

### 3.2. Anatomical Sites Treated

The anatomical sites treated with Mg-based implants varied across the studies. There were no studies that had Mg implants for maxillofaical surgery. All the included studies were on the use of Mg implants for orthopaedic purposes. One study reported usage in the proximal tibia, elbow, upper ankle joint, patella, and distal femur [12]. Two studies focused on the medial malleolus [13,16], while another addressed the femoral head [8]. Scaphoid fractures were discussed in two studies [14,15]. Additionally, one study investigated hallux valgus surgery [17], and another focused on radial head fractures [18].

### 3.3. Outcome Measures of Included Studies

Clinical and radiographic findings were employed to assess the outcomes of using Mg and Ti implants across different anatomical sites (Table 2). The assessment methods included a combination of clinical evaluations such as range of motion, functional outcomes, and various specific scoring systems (e.g., AOFAS), along with radiographic imaging techniques like X-rays and CT scans. The American Orthopedic Foot and Ankle Society Ankle–Hindfoot Scale (AOFAS scale) is a clinical scoring system that combines objective scores from the clinician’s examination and subjective scores of pain and function from the patient [19].

### 3.4. Clinical Outcomes

The clinical outcomes of Mg-based implants were assessed through various parameters, including infection rates, serum level monitoring, and the need for revision surgery (Table 3).

#### 3.4.1. Infection Rate and Serum Level

There was only one study [17] that reported infection, with three cases of superficial cellulitis following implant placement, which resolved after a 1-week course of antibiotics.

Serum magnesium, calcium, and phosphate levels were monitored in two studies to evaluate the systemic impact of magnesium implants [8,16]. No significant changes were observed in these serum levels postoperatively. One study specifically noted that renal function remained stable throughout the postoperative period [16].

#### 3.4.2. Need for Revision

The overall rate of revision surgery across the studies was low, with only 1 case reported out of 275 patients. This single revision surgery was necessary for a patient who experienced a broken pin in the medial femoral condyle due to a highly unstable osteochondral defect [12]. None of the other studies reported any need for revision surgery.

### 3.5. Radiographic Outcome

The radiographic outcomes of magnesium-based implants were evaluated based on several key indicators, including bone union, the time required for bone healing, and the presence of radiolucent zones (Table 3).

#### 3.5.1. Bone Union

Across the studies, bone surgeries with Mg-based implants demonstrated high rates of bone union, with most studies reporting 100% union among their patients. One study reported a slightly lower rate of 98.9%, as there was one patient who experienced a broken pin in the medial femoral condyle due to a highly unstable osteochondral defect [12]. The time required for bone union also varied across the studies. In one study, the time for bone union ranged from 4 to 6 weeks [14]. Another study reported a mean bone union time of 11.2 weeks, with a range of 9 to 14 weeks [15]. The longest reported time for bone union was a mean of 10.2 weeks, with a range from 8 to 16 weeks [18]. In the remaining studies, although specific timelines for bone union were not provided, they reported the success of bone union [8,12,13,16,17]. Overall, the rate of bone union is high, but the time required for bone union is varied.

#### 3.5.2. Radiolucent Zone

Radiolucent zones were observed in seven out of eight of the studies included. The occurrence of these zones was commonly associated with the degradation process of Mg-based implants. In several studies, radiolucent zones first appeared within an average of 2 weeks post-surgery. One study [18] reported the hydrogen gas formation peaked around 8 to 12 weeks, averaging at 8.6 weeks. The time needed for the radiolucent zones to begin decreasing in size varied, with studies reporting a decrease starting around 6 months postoperation [13,16]. For most patients, these radiolucent zones completely disappeared within one year. However, in one study, the radiolucent zones persisted slightly longer, taking up to 18 months to fully disappear [15]. Despite the presence of these zones, the studies consistently reported successful bone union without adverse effects on the overall functional outcomes. Overall, while the occurrence of radiolucent zones is a noted phenomenon, it generally resolves without causing complications.

### 3.6. Comparison with Titanium Implants

Three studies examined the performance of magnesium (Mg) versus titanium (Ti) implants (Table 4).

#### 3.6.1. Rate of Implant Removal

The overall implant removal rate for Ti was 18 out of 111 patients (16.2%) (Figure 2) [13,16,17]. The highest rate of implant removal was reported by Herber et al. [16], with 71% (12 out of 17 patients) requiring removal, primarily due to soft tissue conflict. In the study by May et al. [13], 20% (5 out of 25) of patients with titanium implants underwent removal, with reasons including lateral malleolar nonunion, pain, and difficulty in shoe-wearing. In contrast, Choo et al. [17] reported a much lower removal rate of 1.4% (1 out of 69) for Ti implants; the one case of removal was elective, due to discomfort from implant prominence. Comparatively, the rate of implant removal for Mg implants was low, with 1 out of 275 patients requiring removal (0.4%) [8,12,13,14,15,16,17,18] (Figure 2).

#### 3.6.2. AOFAS Score

The AOFAS scores for Mg implants were slightly higher in [13,17]. For medial malleolar fractures, Mg implants had a score of 93.7 ± 8.8, compared to 90.0 ± 10.7 for Ti implants (*p* = 0.161), indicating no significant difference [13]. Similarly, in the study of hallux valgus surgery, the AOFAS score for Mg implants was 89.5 ± 11.6, which was slightly higher than the 83.6 ± 14.1 recorded for Ti implants (*p* = 0.065), also showing no significant difference [17].

### 3.7. Rate of Mg Implant Degradation (Table 5)

Four studies recorded varying rates of degradation for Mg implants [8,12,16,17]. Herber et al. [16] reported that 90% of patients showed radiographic disappearance of the screw head at one year, although the shafts of the screws remained visible at 6 and 12 months. Stürznickel et al. [12] also reported that one patient had visible degradation of the screw head at one year, and another patient had degradation of the magnesium-based implant after 15 months. Another study observed a decrease in screw diameter by approximately 25% over a 12-month period, with reductions measured at 3.7%, 9.3%, 13.7%, and 25.2% at 1, 3, 6, and 12 months postoperation, respectively [8]. While Choo et al. [17] noticed an almost complete absorption of the screw by one year, Stürznickel et al. [12] pointed out that the combined use of multiple MgYREZr alloy screws within one circumscribed area was safe and did not lead to altered implant degradation characteristics, radiographic findings, or fracture healing. However, follow-up radiography revealed six screws with axial deviation associated with degradation (6/100) [12]. These findings reflect the varying rates of Mg implant degradation for different anatomical sites.

**Table 5 cmtr-18-00024-t005:** Rate of degradation of Mg implants.

Author	Title	Types of Mg Used	Dimension	Anatomical Stie	Rate of Degradation
Stürznickel et al. [12]	Safety and performance of biodegradable magnesium-based implants in children and adolescents	MgYREZr	Diameter: 4.8 mm Length: Not mentioned	Patella	One patient had visible degradation of screw head at 1 year; another patient had degradation of the magnesium-based implants that was observed after 15 months.
Zhao et al. [8]	Vascularized bone grafting fixed by biodegradable magnesium screw for treating osteonecrosis of the femoral head	Pure Mg (99.99 wt.%)	Diameter: 4 mm Length: 40 mm	Femoral head	A decrease in screw diameter of around 25% over a period of 12 months (3.7%, 9.3%, 13.7% and 25.2% reduction at 1, 3, 6, and 12 months postoperatively).
Herber et al. [16]	Can Hardware Removal be Avoided Using Bioresorbable Mg-Zn-Ca Screws After Medial Malleolar Fracture Fixation? Mid-Term Results of a First-In-Human Study	ZX00 rods (99.1 wt% Mg, 0.45 wt% Zn and 0.45 wt% Ca)	Diameter: 3.5 mm Length: 40 mm	Medial malleolar	Of the included patients, 17 (90%) had a radiographic disappearance of the screw head at one year; shafts of screw were still visible after 6 and 12 months.
Choo et al. [17]	Magnesium-based bioabsorbable screw fixation for hallux valgus surgery—A suitable alternative to metallic implants	MgYREZr	Diameter: 3.2 mm Length: Not mentioned	Hallux valgus	A few patients had almost full absorption of the screw which occurred at one year. *

* Author did not specify exact number.

## 4. Discussion

The present systematic review on the potential of bioabsorbable Mg screws in bone surgery presents the following findings: (i) the application of Mg as osteosynthesis material appeared to be limited to orthopedic surgery, and no applications were reported for maxillofacial surgery in the papers included in this systematic review; (ii) low rate of revision surgery (1/275) in Mg group as compared to (18/111) the Ti group, and a low infection rate was found (1/275); (iii) presence of radiolucent zones post-surgery is a common phenomenon and the rate of Mg implant degradation varies in different anatomical sites; (iv) high rate of bone union was found in the Mg group; (v) there did not seem to be an effect on the serum level of Mg and Ca.

Recent developments in material science have significantly influenced the field of bone fixation. Traditionally, Ti has been identified to be the “gold standard” due to its excellent mechanical properties, biocompatibility, and ability to provide stable fixation [20]. Its widespread use is supported by its high success rates, making it a reliable choice for various surgical applications [20]. The comparative performance of Mg and Ti implants in orthopedic surgery has been the focus of numerous studies, many of which have highlighted the potential advantages of Mg implants [12,13,16,17,21,22,23]. Mg implants have demonstrated similar outcomes in terms of patient satisfaction and clinical effectiveness. The studies included in our review have shown that the clinical outcomes measured by the AOFAS scores are comparable between Mg and Ti implants [13,17]. A meta-analysis conducted by Fu et al. found no significant difference in postoperative implant fracture rates (*p* = 0.51) and infection rates (*p* = 1.00) between Mg and Ti implants in the treatment of hallux valgus with distal metatarsal osteotomies [24]. Meanwhile, Plaass et al. [25] reported a higher rate of implant removal in the titanium group than in the magnesium group for distal metatarsal osteotomies. The higher rate of Ti implant removal is mainly due to tissue irritation, pain, and implant prominence. Other studies also reported similar findings [16,25].

Despite the claimed high success rate by some studies, instances of implant failure have been reported in clinical applications. For example, in a study by Könneker et al. [26], it was observed that, if bioabsorbable Mg screws protrude and cause symptoms, they must be removed to prevent damage to adjacent cartilage and vital structures before they fully dissolve. This emphasizes the importance of monitoring the placement and behavior of Mg implants closely during the postoperative period. In another study, Ünal et al. [27] mentioned that implant failure can be determined by a combination of radiographic and clinical findings. The breakage of a Mg screw before bone healing could be seen as a failure, while breakage after successful bone healing might represent the normal degradation process of the implant [27]. As Mg-based implants degrade, their mechanical stability naturally diminishes, potentially leading to axial deviation. However, if axial deviation occurs without accompanying signs of implant degradation and results in the non-healing of the surgical site, revision surgery may be necessary [12].

The rate of implant removal in the Mg group remained low and it can be explained by their biocompatible nature. Radiolucent zones were frequently observed in the included studies, and they were closely associated with the degradation process of these materials [12,13,14,15,16,17,18]. Yet, studies consistently reported successful bone union without adverse effects on the overall functional outcomes [13,14,15,16,17,18,28]. The mechanism of the formation of radiolucent zones can be explained by the corrosion of Mg in an aqueous environment, which occurs through an electrochemical reaction, where Mg reacts with water to produce magnesium hydroxide (Mg(OH)_2_) and hydrogen gas (H_2_) [29]. In vitro studies suggest a high volume of hydrogen gas production (approximately 1 mL/mg of Mg) [30]. While these zones generally begin to diminish in size after six months and are usually resolved within a year [13,14,16,17,31], they may persist up to 18 months in some instances [15]. Therefore, it is crucial to recognize that the presence of radiolucent zones is a normal occurrence and should not be mistaken for osteolysis or cyst formation. The Hounsfield Units of CT scans may be a valuable method that can be used for differentiation [32]. Meier and Panzica [33] reported on five patients with scaphoid fractures treated with magnesium screw fixation; here, early postoperative radiographic findings aligned with those of cyst formation in three cases. However, follow-up radiographs confirmed that bone union was eventually achieved in all patients, and functional outcomes were excellent.

There was a varied time of radiolucency disappearance among the included studies. Various factors such as implant size, coating, and anatomical location can affect the rate of reduction in size of radiolucent zones. For example, Delsmann et al. [34] have shown that 2.7 mm and 3.2 mm Mg screws typically resorb within 100 to 200 days, whereas ceramic-coated 4.8 mm screws exhibit a slower reduction in radiolucent zones, taking approximately 400 days to fully disappear. The larger surface area of these screws may contribute to more extensive radiolucent zones and a prolonged resorption period. The ceramic coating also delays the initial corrosion process during the first month [34]. In terms of anatomical site, Zhao et al. [8] noted that no gas accumulation was observed in hip joints, likely due to the larger tissue space, which reduces the likelihood of gas buildup. Stürznickel [12] and Herber et al. [16] reported visible degradation of the screw head at one year; possible explanations of such a phenomenon include shear forces present in the soft tissue and the abundant blood supply adjacent to the screw head that might lead to accelerated degradation. This can be an additional advantage as it prevents soft tissue irritation associated with Ti implants. An in vitro study found that the different environment pH levels will affect the degradation rate of Mg. When pH exceeds 11.5 in a Hank’s solution (simulated body fluid), stable magnesium hydroxide is produced, forming a protective layer on the surface that reduces implant corrosion. In contrast, in acidic environments, the corrosion rate exceeds 800 μm per day at pH 5.5 when compared to 6 μm at pH 7.4 [35]. Given that the pH surrounding the screw will drop postoperatively, the degradation rate of Mg would therefore be higher [13]. However, two of the included studies only observed the degradation of screw heads at one year [12,16]. This can be explained by the advances in material science and the addition of alloy to reduce and control corrosion rates. Stürznickel [12] and Herber [16] used MgYREZr screws and ZX00 rods, respectively; yttrium and zinc have been shown to have an ability to delay the degradation rates of Mg screws in an in vitro study [30]. In clinical practice, different strategies such as alloying and surface modification were developed to ensure that the degradation rate is slow enough to allow optimal bone healing [36].

The degradation products of Mg screws have been used to demonstrate the ability of Mg to stimulate osseous growth [37]. Clinical effectiveness in terms of the bone healing of Mg-based screws is well supported by multiple studies, indicating their potential as a reliable alternative to titanium screws in orthopedic surgery. Fu et al. conducted a meta-analysis comparing the fixation of distal chevron osteotomy (DCO) using Mg and Ti screws [24]. While both types of screws were found to provide equal stabilization, the results proved the biomechanical sufficiency of magnesium screws for such procedures [24]. Similarly, Ünal et al. [27] reported on the use of 4.8 mm bioabsorbable Mg screws for tibial tubercle osteotomy (TTO), in which these screws maintained sufficient fixation without displacement or failure until bone union was achieved. They concluded that Mg screws are not only clinically satisfactory but also biomechanically capable of withstanding the physiological loads encountered during knee extension. In addition to these findings, Plaass et al. [32] highlighted the anti-infective properties of magnesium due to its high pH during degradation, with bone trabeculae bridging typically observed within the first six weeks postoperation. Our systematic review further supports these results, demonstrating consistently high bone union rates across studies using magnesium implants, with almost all studies reporting 100% union among patients [8,12,13,14,15,16,17,18]. Only one study noted a slightly lower rate of 98.9% due to a broken pin in the medial femoral condyle [12]. The time required for bone union varied from 4 to 16 weeks, indicating that magnesium screws offer both effective stabilization and reliable bone healing.

The biocompatibility of Mg implants is a critical aspect of their application in orthopedic surgery, with a strong emphasis on patient safety and potential angiogenesis benefits. Mg implants have shown promising biocompatibility, as evidenced by low infection rates and stable serum levels [8,12,13,14,15,16,17,18]. In our systematic review, the findings suggest that renal function stayed stable throughout the postoperative period; serum levels of magnesium, calcium, and phosphate were found to remain within normal physiological ranges, suggesting that the degradation products of Mg implants do not adversely affect metabolic balance [8,16]. Despite these positive outcomes, there are concerns regarding the potential development of hypermagnesemia following the use of Mg-based implants; this is particularly highlighted in in vivo studies [38,39]. Hypermagnesemia, characterized by elevated serum magnesium levels, can lead to symptoms such as hypotension, nausea, mental impairment, and, in severe cases, neuromuscular and cardiovascular dysfunction, which could result in respiratory failure or cardiac arrest [39]. To our knowledge, a limited number of human studies carried out blood tests to monitor serum levels and they reported stable magnesium serum levels postoperatively [16,40,41]. Our systematic review results found that no significant changes had been reported in post-implantation serum magnesium levels, suggesting that the risk of hypermagnesemia in clinical practice may be low [16].

Magnesium-based implants have shown versatility across different anatomical sites, including the tibia, elbow, ankle, patella, and femur, as well as in more specialized areas such as scaphoid fractures and hallux valgus surgery [42]. The trend toward using magnesium in bone fixation in different anatomical sites is steadily increasing. Clinical studies have demonstrated that Mg-based implants, such as biodegradable magnesium alloy screws, are effective in the fixation of mandibular condylar fractures. Leonhardt et al. [43] reported satisfactory results with these implants, noting that patients experienced no facial nerve palsy and achieved satisfactory occlusion, with no restrictions in mandibular function within three months post-surgery. Additionally, a study investigating the use of Mg-based biodegradable headless compression screws for mandibular condylar fractures showed excellent clinical outcomes; all patients in this study exhibited well-restored temporomandibular joint (TMJ) function, significant improvement in mandibular function and occlusion, and no need for implant removal or revision surgery [44]. Radiographic outcomes also supported the efficacy of Mg-based implants, with CBCT scans showing no screw breakage, successful bone healing, and complete filling of fracture gaps with new bone tissue within 12 months postoperatively. To our knowledge, most studies on magnesium screws have primarily focused on maxillofacial trauma rather than orthognathic surgery. Conversely, research on co-polymers has predominantly centered on orthognathic surgery, which provides insights into the effectiveness of magnesium implants, considering that magnesium implants provide better mechanical stability and biocompatibility. A meta-analysis conducted by Al-Moraissi et al. [45] reported there was a statistically higher rate of intraoperative fracture of plates and screws in the biodegradable group of orthognathic surgery when compared to their conventional counterparts. Studies by Cheung et al. [46], Ueki et al. [47], and Choi et al. [48], in which they studied Le Fort l surgeries using PLLA screws and plates, reported that plate fixation showed significant upward displacement in the anterior maxilla, and it did not provide enough vertical stability in the maxillary position due to muscle forces. There were no significant differences between skeletal class ll and lll patients at 1 year postoperatively, suggesting that the preoperative skeletal class and occlusion did not affect bone healing after Le Fort l surgery. On the contrary, Ti plate and screw fixation have been in practice for a long time in the field of maxillofacial surgery; these approaches are commonly used in open reduction internal fixation (ORIF) of mandibular condylar fractures and orthognathic surgery. Despite the fact that plate removal rates remain low, these plates still require removal when certain complications occur, such as unacceptable palpability [49], infection [50], and screw tip perforation of the condylar surface, which can occur during the bone remodeling process or under the strong functional load of the orofacial system [43].

Therefore, long-term clinical studies are required to draw a more definite conclusion on whether Mg screws can minimize postoperative complications and perform in the field of maxillofacial surgeries. Future research into the application of magnesium (Mg) implants in bone fixation in different anatomical sites is essential to fully harness their potential benefits for patients. In addition to clinical efficacy, a cost–benefit analysis comparing Mg implants to traditional titanium implants should be conducted, taking into account the potential for reduction in revision surgery rates. Furthermore, optimizing the corrosion rate of Mg implants through advancements in alloying techniques, surface treatments, and coating technologies will be key to enhancing their performance and expanding their clinical use. The application of Mg in orthognathic surgery and other bone fixation procedures offers significant potential; however, such applications require careful evaluation of clinical and radiographic outcomes to fully utilize their advantages.

## 5. Limitations

A potential limitation of the current review is that different types of magnesium implants were used in the different included studies, and they were placed in different anatomical sites. Taking the understanding that each implant type may have different clinical performances, the papers included here did not provide enough information to analyze each individually. The follow-up time was set at 1 year for papers to be included in this review, but a longer follow-up period and long-term clinical studies are crucial for understanding the overall effectiveness and rate of degradation of Mg implants.

## 6. Conclusions

The present systematic review investigated the safety of bioabsorbable magnesium-based material applications for bone fixation in humans. The evidence suggested that (i) magnesium implants provided a favorable safety profiles and are biocompatible with the surrounding tissues in terms of clinical outcome, rate of revision surgery, serum mineral level, and infection rate. The review also suggested that (ii) the presence of radiolucent zones post-surgery is common and safe. (iii) Bone union rate was found to be high following the use of magnesium implants. This study also pointed out that (iv) magnesium implants may perform better than their titanium counterparts, as the magnesium implants were found to yield a lower rate of implant removal. Despite the fact that applications of Mg-based materials in craniomaxillofacial surgery are limited so far, the material’s high biocompatibility, mechanical reliability, and proven safety could mean that Mg-based materials are promising; they provide prospective applications as new biodegradable osteosynthesis materials with improved clinical outcomes.

## Figures and Tables

**Figure 1 cmtr-18-00024-f001:**
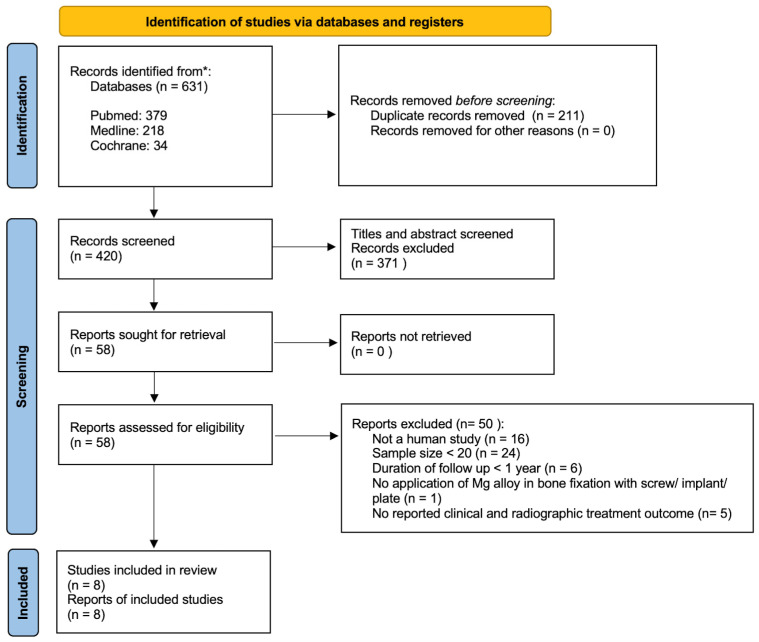
PRISMA flow diagram for article selection (adopted from the PRISMA 2020 flow diagram).

**Figure 2 cmtr-18-00024-f002:**
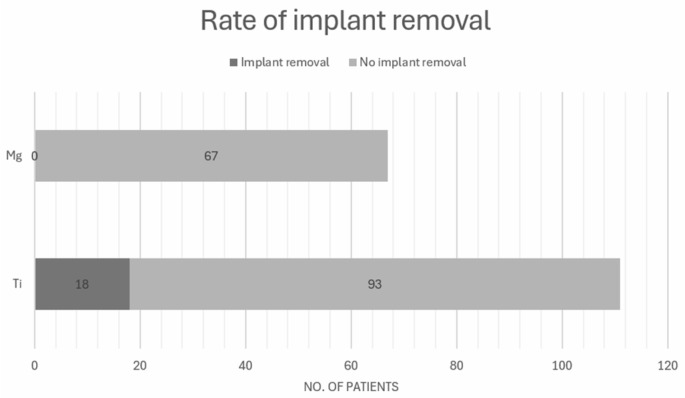
Magnesium and titanium rate of implant removal.

**Table 3 cmtr-18-00024-t003:** Clinical outcomes of Mg-based implants of the included studies.

Author	Title	Type of Mg Used	Anatomical Site	Infection	Serum Level	Revision Surgery Performed (Per Patient)	Bone Union	Time for Bone Union (Weeks)	Presence of Radiolucent Zones
Stürznickel et al. [12]	Safety and performance of biodegradable magnesium-based implants in children and adolescents	MgYREZr	Proximal tibia; elbow; upper ankle joint; patella; distal femur	0/89 (0%)	Not mentioned	1/89	88/89 (98.9%) patients achieved bone union	N/A	All patients 89/89 and decrease in size during follow-up examinations (mean follow-up duration was 8.2 months).
May et al. [13]	Bioabsorbable magnesium screw versus conventional titanium screw fixation for medial malleolar fractures	MgYREZr	Medial malleolar	0/23 (0%)	Not mentioned	0/23	All patients (23/23) (100%) achieved bone union	N/A	All patients (23/23), the amount of gas increased in the first 4 months and began to decrease after the 6th month and disappeared within 1 year.
Zhao et al. [8]	Vascularized bone grafting fixed by biodegradable magnesium screw for treating osteonecrosis of the femoral head	Pure Mg (99.99 wt.%)	Femoral head	0/23 (0%)	No significant changes of serum Ca, Mg, and P at 1, 7, and 14 days postoperation	0/23	Mg screw promoted bone mineral density	N/A	Not observed.
Lee JW et al. [14]	Long-term clinical study and multiscale analysis of in vivo biodegradation mechanism of Mg alloy	5 wt% Ca, 1 wt% Zn screw	Scaphoid, distal radius	0/53 (0%)	Not mentioned	0/53	All patients (53/53) (100%) achieved bone union	Range: 4–6	After 6 months of surgical fixation, the distal radius fracture was completely healed with a small radiolucent area; completely healed in 1 year.
Polat et al. [15]	Surgical outcomes of scaphoid fracture osteosynthesis with magnesium screws	MgYREZr	Scaphoid	0/21 (0%)	Not mentioned	0/21	All patients (21/21) (100%) achieved bone union	Mean: 11.2 (range: 9–14)	All patients (21/21), radiolucent zone decreased in size at 9 months and completely disappeared within 18 months.
Herber et al. [16]	Can Hardware Removal be Avoided Using Bioresorbable Mg-Zn-Ca Screws After Medial Malleolar Fracture Fixation? Mid-Term Results of a First-In-Human Study	ZX00 rods (99.1 wt% Mg, 0.45 wt% Zn, and 0.45 wt% Ca)	Medial malleolar	0/20 (0%)	Normal levels of Mg and Ca; renal function remained stable	0/20	All patients (20/20) (100%) achieved bone union	N/A	19/19 had a radiolucent zone noted, slightly decreased from 6 months to 1 year.
Choo et al. [17]	Magnesium-based bioabsorbable screw fixation for hallux valgus surgery—A suitable alternative to metallic implants	MgYREZr	Hallux Vagus	3/24 (12.5%)	Not mentioned	0/24	All patients (24/24) (100%) achieved bone union	N/A	Soft tissue gas shadows were observed on 3-month X-rays, all of which were resolved on the 12-month postoperative X-ray.
Lee CH et al. [18]	Results of the Use of Bioabsorbable Magnesium Screws for Surgical Treatment of Mason Type II Radial Head Fractures	Mg Screw (Resomet:U&I Corp Seoul, Republic of Korea)	Radial head	0/22 (0%)	Not mentioned	0/22	All patients (22/22) (100%) achieved bone union	Mean: 10.2 (range 8–16)	First appeared at an average of 2 weeks after the surgery, peaked at an average of 8.6 weeks (range, 8–12 weeks), and gradually absorbed and disappeared.

**Table 4 cmtr-18-00024-t004:** Comparisons of the clinical outcomes between Mg versus Ti implants in the included studies.

Author	Title	Type of Ti Used	Types of Mg Used	Anatomical Site	Mg rate of Implant Removal (Per Patient)	Ti Rate of Implant Removal (Per Patient)	AOFAS Ti (1 Year Post-op)	AOFAS Mg (1 Year Post-op)	*p* Value
May et al. [13]	Bioabsorable magnesium screw versus conventional titanium screw fixation for medial malleolar fractures	Conventional titanium screw	MgYREZr	Medial malleolar	0/23 (0%)	5/25 (20%)	90.0 ± 10.7	93.7 ± 8.8	0.161
Herber et al. [16]	Can Hardware Removal be Avoided Using Bioresorbable Mg-Zn-Ca Screws After Medial Malleolar Fracture Fixation? Mid-Term Results of a First-In-Human Study	Conventional titanium screw	ZX00 rods (99.1 wt% Mg, 0.45 wt% Zn and 0.45 wt% Ca)	Medial malleolar	0/20 (0%)	12/17 (71%)	/	89.8 ± 7.1	/
Choo et al. [17]	Magnesium-based bioabsorbable screw fixation for hallux valgus surgery—A suitable alternative to metallic implants	Conventional titanium screw	MgYREZr	Hallux Valgus	0/24 (0%)	1/69 (1.4%)	83.6 ± 14.1	89.5 ± 11.6	0.065

## Data Availability

The raw data supporting the conclusions of this article will be made available by the authors on request.

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
