# Peer review of "Bioabsorbable Magnesium-Based Materials Potential and Safety in Bone Surgery: A Systematic Review"

_1943-3883, 2025, doi:10.3390/cmtr18020024_

Round 1

Reviewer 1 Report

Comments and Suggestions for Authors

Dear authors,

I gladly reviewed your manuscript, and I would welcome your response to my comments and recommendations.

This is one of the best prepared manuscripts I have reviewed this year. There is not much room for improvement from my point of view. However, I was wondering why several studies did not end up in your review. For example, while reviewing related literature I came upon this study (https://link.springer.com/article/10.1007/s00247-022-05383-x) and it is not obvious to me why it was excluded.

In addition, the authors cite the meta-analysis by Fu X et al. The reason for exclusion of the studies include in that meta-analysis are not apparent to my either.

Since there is a strong overlap and redundancy between tables 2, 3, 4 and 5. I suggest merging the tables and reducing their number to two.  I don’t think the title is important in the table. The table can be printed on a landscape positioned page. Please replace City names with China/Hong Kong.

The risk of bias in the included studies is currently shown in the methods section, but it belongs in the results section.

Figure 2. The unit of the X-axis is not clear. The quality of the figure can be improved.

Kind regards

Reviewer 2 Report

Comments and Suggestions for Authors

The manuscript has a very relevant theme for the area of Maxillofacial Surgery, interesting to be explored and to provide new research and scientific studies in the area of Maxillofacial Surgery, just as it has been developed in orthopedics. I suggest publishing the manuscript, as it presents a good development of the methodology, writing and exposition of the results available in the literature, as well as the discussion of the favorable results found. Its publication will be a great contribution to the academic community for future studies on the use and feasibility of using magnesium screws, especially in bone graft surgeries for dental implants and major maxillofacial reconstruction.

Reviewer 3 Report

Comments and Suggestions for Authors

Dear authors, thank you for submitting this paper. It can bring attention to the use of Mg based materials in OMFS. 

Reviewer 4 Report

Comments and Suggestions for Authors

Dear Authors,

After a detailed analysis of the submitted manuscript, several discrepancies were identified in the methods and results, as well as some limitations that compromise the quality of your systematic review.

Issues in the Methods

A systematic review must follow rigorous and well-documented methods in accordance with PRISMA, the Cochrane Handbook, and other relevant guidelines to ensure transparency and reproducibility.

Search Strategy

  • The manuscript mentions an initial search in electronic databases but does not specify which databases were used. For systematic reviews, it is recommended to search multiple relevant databases.
  • There is no mention of a search for grey literature, such as theses, conference proceedings, and clinical trial registries, which may introduce publication bias.
  • The inclusion and exclusion criteria are not clearly detailed, making it difficult to assess study eligibility.
  • There is no clear justification for article exclusions. A table listing excluded studies would enhance transparency.
  • The manuscript states that a tool was used to assess the risk of bias in the included studies. However, ROBINS-I is designed for observational studies. In the results section, the authors included an RCT, six case series, and a cohort study. For randomized controlled trials (RCTs), the RoB 2 tool should be used. How can this review be considered systematic under these circumstances?
  • There is no description of the quality assessment of the included studies, making it difficult to determine whether the conclusions are based on robust evidence or lower-quality studies.

Issues in the Results

The results of a systematic review should be presented clearly and in a structured manner. However, several issues were identified:

  • There is no analysis of heterogeneity, which is essential to assess the compatibility of results across different studies.

Issues in Data Presentation

  • The data are presented in a superficial manner, with generic statements such as "high bone union rate in the Mg group" without providing exact figures or confidence intervals.
  • There is no mention of confidence intervals or p-values, which compromises the statistical interpretation of the findings.

Study Limitations

All systematic reviews should explicitly discuss their limitations. The main limitations identified include:

  • Potential publication bias.
  • Low generalizability (The manuscript suggests that magnesium implants may be a safe alternative but does not discuss the limitations of the primary studies, such as small sample sizes and variability in measured outcomes).
  • Absence of a subgroup analysis.

Considering these issues, further improvements are necessary to enhance the rigor and transparency of the review.

Reviewer 5 Report

Comments and Suggestions for Authors

Review of:

Bioabsorbable magnesium-based materials potential and safety in bone
surgery: a systematic review
Authors: Chun Ho Hung, Yui  Chit Kwok, Jason Yip, Ho Hin Wong, Yiu Yan Leung

Manuscript ID: cmtr-3378389

The overall premise for this manuscript is both interesting and has a practical application

The commendable parts to this submission are:

A systematic review of a relevant area of clinical practice

Conducted according to the PRISMA guidelines

Clear study design, selection criteria, search strategies used, risk of bias considered, analysis of some relevant articles performed, some relevant clinical outcomes considered, analysis performed - but seems to miss the main thrust of the article in terms of OMFS applicability

Assessed that Magnesium implants seem to have a good safety profile

The extremely limited literature in relation to use of magnesium implants for OMFS use in Humans is considered (but not the dental use of such)

Suggestion that a cost:benefit analysis study would be of use (especially in government funded services)

The conclusion is that Magnesium materials could be a promising prospect (in Cranio/OMFS applications) which is true, but does not add to our current literature or understanding and specifically the applicability on this subject in particular

The critical points in this submission are as follows, that require further consideration:

Overall this manuscript misses the most relevant target audience, which is Orthopaedics. The evidence presented is for use of such implants ONLY in Orthopaedic patients, there are no significant human studies on the use of such implants for OMFS applications. Submission to CMTR therefore does not seem to be relevant

There is no explanation as to how applications and uses in Orthopaedics can be translated into use in OMFS applications and in Humans

Further, there is no convincing argument used or explained as to why we actually need an alternative to the current use of biocompatible Titanium implants in OMFS applications, in Humans. Is there such a significant problem with infection/plate exposure/palpability/pain/poor wound or fracture healing/malocclusion or other adverse events that we need to consider an alternative? Where is the convincing evidence of this? It seems that this ground has been trodden before with multiple studies on the use of PLLA, etc. materials in OMFS applications and these studies often concluded that the resorbable plate alternatives were bulky, led to infections/ reactions/ complications – so why are we considering a resorbable alternative again? and what are the advantages of Magnesium based resorbable materials over the PLLA type previously used?

Also, there needs to be consideration of different practices for example in the UK and other parts of the world – what are the reasons oft quoted for the removal of Titanium hardware used in OMFS applications in different countries? This is highly relevant because in some countries the reasons will be driven by clinical outcomes e.g. plate exposure/infection/symptoms i.e. potentially an emergency situation, whereas in other countries there may be other financial incentives to remove such hardware, which is then performed on a purely elective basis

It is remarkable that only 8 articles were finally selected for their study, from the initial 420 after duplicates removed, which is a small number on which to base conclusions. Also, given that there are no significant and scientifically robust human studies (barring two small case series on the use of headless screws in TMJ/condylar fractures) on the use of Magnesium implants for OMFS applications, it is therefore no surprise that their search strategy revealed basically no relevant articles, as it also excluded relevant animal studies on this research

The studies on headless screw use for TMJ/condylar fracture fixation are not representative of the bulk of OMFS applications and in particular their use in this context is in an anatomical location away from the oral cavity, which is a crucial consideration in most OMFS applications

Specifically, the effect of saliva, exposure to paranasal sinus lining, occlusal forces or the effect of TMJ movements on Magnesium based materials and the use of such materials in different maxillofacial and craniofacial bone areas in humans (with attendant lines of tension and compression) and situations which may require thicker plates/locking and load bearing fixation, must all be studied and considered prior to their recommendation. Along with the degradation process and tissue response to such materials in Humans

The most significant study which summarises animal studies on the use of Magnesium implants in OMFS applications is as follows and interestingly this is also in a MDPI journal:

Applications of Biodegradable Magnesium-Based Materials in Reconstructive Oral and Maxillofacial Surgery: A Review. Sanja Vujovi´c, Jana Desnica, Dragana Staniši´c, Irena Ognjanovi´c, Momir Stevanovic,  Gvozden Rosic. Molecules 2022, 27, 5529. https://doi.org/10.3390/molecules27175529.

So, whilst there have been other isolated animal studies on the use of Magnesium implants for OMFS applications in other OMFS journals over the last few years, it remains that the above article is the most definitive currently in outlining where the scientific community is at the present time in relation to such studies. It therefore seems most apposite to suggest that an “expert review/opinion article” is now required in a leading OMFS journal which summarises the relevant research to date on this topic, that outlines the actual need to consider replacement of Titanium implants for OMFS use (be that clinical/financial/other) and indicates how we can move from the animal/laboratory phase of evaluation to actual human studies, so as to assess their applicability/impact/results/utility both from clinical and radiographic points of view in the human environment prior to standard clinical use

Other weaknesses identified are as follows: even though the PRISMA guidance was followed, I am unable to identify if PROSPERO registration (or another registration alternative for systematic reviews) happened and yet this is recommended. Other issues with this study are that studies are considered with the use of Magnesium based material of different composition e.g. alloys, there is no specific time frame at which the serum level for Magnesium was assessed which could be relevant or any explanation as to why there seem to be no changes in serum Magnesium level post-operatively , in the articles chosen there is missing data and variability in the measurement of outcomes – which affects the risk of bias (50% of the articles selected are low risk but the other 50% are at moderate risk – table 1), there is one study on a Paediatric population – which could conceivably respond differently to Magnesium implants compared to an adult population, all of the selected articles barring one are case series (not randomised controlled trials) which clearly reduces the reliability of the outcomes/results, how do the times for bone union in table 5 for the Magnesium based materials compare to the times for bone union with Titanium based materials? Interestingly, if removal of the (Titanium implant) is the driver for a move to Magnesium based materials – it is of note that even such materials might also still require removal (table 6), even though there is a description of magnesium based material decomposition with Hydrogen gas production – there is no assurance provided that there have not been any cases of significant gas embolism

Comments on the Quality of English Language

Overall good English

Round 2

Reviewer 4 Report

Comments and Suggestions for Authors

After the responses provided by the authors, I consider the article suitable for publication.

Author Response

We thank Reviewer 4 for the comments and time

Reviewer 5 Report

Comments and Suggestions for Authors

The problems, as previously identified remain with this systematic review, namely:

  1. Whilst the review has been done according to protocol, the only articles identified involved Orthopaedic cases, not craniomxillofacial cases
  2. It is not possible to translate findings in Orthopaedic studies to craniomaxillofacial applications, because both applications/environments e.g. saliva in the oral cavity and uses, plus biomechanics, are critically different
  3. The most relevant evidence to date for Mg implants in CMF applications is in animal studies only (of which there are several in the CMF literature already), which cannot be directly translated to humans
  4. There is only 1 study using Mg pins for condylar fractures in humans - which had a very low study number and did not have direct translation to all of the other CMF applications as it was only for TMJ/condylar head fractures
  5. Studies on PLLA implants have been done in humans, but PLLA is a different material to Mg, further PLLA implants have not significantly replaced the use of Titanium in CMF surgery
  6. Overall, this systematic review does not add any significant clinical value to current CMF practice, it is more applicable to Orthopaedic practice and the evidence base cannot be changed until there are more studies on the use of Mg implants in humans be they randomised controlled trials/case control series or otherwise
  7. There is also an evidence base for the use of Mg implants/resorbable meshes in Implant Dentistry in Humans, which is not covered at all in this systematic review, but could have been considered
Comments on the Quality of English Language

Acceptable

Author Response

We thank Reviewer #5 for the time and comments. Our replies to the comments areas follow:

Q: Whilst the review has been done according to protocol, the only articles identified involved Orthopaedic cases, not craniomxillofacial cases

Response: We appreciate Reviewer 5’s time and constructive feedback. This systematic review aims to enhance surgeons’ understanding of Mg in fixation and its biological behaviour, with potential applications in craniomaxillofacial surgery. We believe our findings help bridge a critical knowledge gap in this field and encourage further clinical translation

Q: It is not possible to translate findings in Orthopaedic studies to craniomaxillofacial applications, because both applications/environments e.g. saliva in the oral cavity and uses, plus biomechanics, are critically different

Response: We respect the comments from Reviewer 5. However, we believe the application of bone fixation has many similarities in the application of orthopaedics and the craniomaxillofacial bones. In fact, from history, several procedures of craniomaxillofacial surgeries, including bone fixation, were first used in orthopaedic surgery. We believe having an insight from the use of Mg fixation from orthopaedic has a significant value for its potential use in the craniomaxillofacial region.

Q: The most relevant evidence to date for Mg implants in CMF applications is in animal studies only (of which there are several in the CMF literature already), which cannot be directly translated to humans

There is only 1 study using Mg pins for condylar fractures in humans - which had a very low study number and did not have direct translation to all of the other CMF applications as it was only for TMJ/condylar head fractures

Response: We would like to clarify that the inclusion criteria for the final review in this systematic review were (i) clinical study on human subjects, (ii) consisted of a sample size with at least 20 subjects, (iii) reported the duration of subjects follow-up, at least 1 year follow up (iv) ap-plication of Mg in bone fixation with implant/ screw/ plate (v) reported clinical and radio-graphic treatment outcome. Articles that satisfied all five criteria were selected for the final review. Studies that did not fulfil these criteria were excluded.

Q: Studies on PLLA implants have been done in humans, but PLLA is a different material to Mg, further PLLA implants have not significantly replaced the use of Titanium in CMF surgery

Response: Our systematic review focused in evaluating the clinical outcome, safety and applications of Mg-based osteosynthesis material. PLLA is not the subject of our systematic review.

Q: Overall, this systematic review does not add any significant clinical value to current CMF practice, it is more applicable to Orthopaedic practice and the evidence base cannot be changed until there are more studies on the use of Mg implants in humans be they randomised controlled trials/case control series or otherwise

Response: While we respective the comment of Reviewer 5, we would like to express once again this study was to enhance surgeons’ understanding of Mg in fixation and its biological behaviour, with the chance to explore the potential applications in craniomaxillofacial surgery.

Q: There is also an evidence base for the use of Mg implants/resorbable meshes in Implant Dentistry in Humans, which is not covered at all in this systematic review, but could have been considered

Response: We respect Reviewer 5 for mentioning the use of Mg meshes for implant dentistry. The aim of the systematic review was to evaluate the clinical outcome, safety and applications of Mg-based osteosynthesis material. Since the applications for dental implants or bone grafting are completely different from the biological and biomechanical needs for osteosynthesis, it is not within the scope of this systematic review.